# Ecosystem Spatial Changes and Driving Forces in the Bohai Coastal Zone

**DOI:** 10.3390/ijerph16040536

**Published:** 2019-02-13

**Authors:** Min Cheng, Binbin Huang, Lingqiao Kong, Zhiyun Ouyang

**Affiliations:** 1State Key Laboratory for Urban and Regional Ecological, Research Center for Eco-environmental Sciences, Chinese Academy of Sciences, Haidian District, Beijing 100085, China; cmin611@163.com (M.C.); 17190196351@163.com (B.H.); lqkong@rcees.ac.cn (L.K.); 2Research Center for Eco-environment Sciences, University of Chinese Academy Sciences, Beijing 100049, China

**Keywords:** ecosystem spatial changes, land use transition matrix, landscape metrics, driving forces

## Abstract

Landscape change is an important aspect of coastal ecological conservation and has an essential influence on the sustainable development of the coastal economy. With remoting-sensing (RS) images between 2000, 2005, 2010, and 2015, using geographic information system (GIS) technologies, we examined ecosystem spatial changes in the Bohai coastal zone. Results showed that wetlands, mainly constituted by reservoirs/ponds, were the dominant landscape types. The urban ecosystem has the largest area increment and the fastest growth rate from 2000 to 2015. The quantification of landscape metrics revealed that spatial patterns have changed significantly, and the change direction of these ecosystems had moved toward increased heterogeneity and fragmentation. In addition, natural and socio-economic data were used to analyze the major driving forces triggering ecosystem spatial changes through redundancy analysis (RDA). The results revealed that the output of aquatic products (AQ) and population (Pop) were the main factors related to wetland ecosystem change. Pop and gross domestic product per capita (GDPpc) were closely related to the urban ecosystem change. Annual mean temperature (ATm), crop acreage (CA), and grain yield (GY) had positive correlations with the agriculture ecosystem changes.

## 1. Introduction

Coastal zones are significant ecological boundaries, forming the transition area between terrestrial and marine ecosystems [1]. Coastal ecosystems provide easily accessible goods and services to humankind and play a significant role in coastal economic development and political interactions between countries [2]. One third of the world’s population living within 100 km of a coast [3], and coastal communities are nearly three times more densely populated than inland areas [4]. The coastal region of China is comprised of an area of more than 3 million km^2^ and possesses an 18,000 km coastline stretching across tropical, subtropical, and temperate zones [5]. It is estimated that more than 70% of large Chinese cities are located in coastal zones, and coastal development plays a dominant role in the national economy. The value of coastal ecosystems accounts for more than 50% of its gross domestic product (GDP) [6]. The long-term sustainable development of coastal communities and the quality of human life closely depend on coastal ecosystems and the crucial services they generate, such as fishery production, climate mitigation, storm buffering and waste treatment [4]. The quality and quantity of ecosystem services are directly affected by coastal land cover/use change [7,8,9]. However, as the most densely populated area, Bohai coastal regions have experienced continuous alteration and transformation over the past decades, from wetland, forest and grassland to farmland and urban. These changes have negatively affected the ecosystem composition and structure, altered the production capacity and transformed the ecological attributes of the ecosystems, and influenced the nutritional transport between soil and vegetation [10]. Some natural coastal ecosystems are being continuously altered, transformed or destroyed which has resulted in the degraded function of ecosystem services, including ecosystem goods and services provision, environmental pollution control, biodiversity conservation, and human vulnerability to changing ecosystems [11,12]. Quantitative information on the historic change of landscape structure and composition was helpful for understanding the consequences of landscape changes [13,14]. Landscape pattern change analysis is increasingly considered an effective way for facilitating better policy decision-making in the sustainable development of coastal management. Landscape pattern change is usually quantified by landscape metrics [15,16,17]. Landscape pattern metrics are simple quantitative indices that concentrate landscape information and reflect the change characteristics of the ecosystem structure and spatial configuration. In recent decades, numerous researchers have focused on landscape status, landscape evolution, forecasting future landscape change, etc. [18,19,20]. Based on the reconstruction of historical landscape information, long-term human-environment interactions can be analyzed to develop a comprehensive understanding of these changes [21].

Driven by endogenous and exogenous factors in different spatial and temporal scales, the landscape is always in a state of dynamic change. Recent research has focused on the reasons behind the landscape changes and the “driving forces” have been developed into a fundamental concept [22], which is now used as a framework for identifying the causes, processes and consequences of landscape changes and has become indispensable for the assessment of policy decisions [23]. The definition of driving forces is the forces that cause observed landscape changes [24]. There are five major types of driving forces: socioeconomic, political, technological, natural and cultural driving forces [24]. These driving forces are not independent, and they influence the landscape change through non-linear interactions. The analysis on the driving forces of landscape change is a sort of directional research, and therefore there is no specific method or framework. However, statistical analysis is helpful to identify correlations between landscape changes and driving forces. Many studies analyzed the driving forces from the perspective of nature and socio-economic development [25,26]. The analysis of landscape spatial changes and its driving forces in different periods has become the latest trend. Although most current achievements have not explained fully the casual relationship and driving mechanism between factors and changes, they lay a good foundation for the further research in this field.

This study evaluated landscape changes of the coastal ecosystems in Bohai Bay from 2000 to 2015. With landscape pattern changes in Bohai Bay, especially the urban expansion and shrinkage of wetland ecosystems, certain ecological problems have emerged, such as wetland degradation and habitat loss and fragmentation. The quantitative analysis of landscape changes is necessary for the settlement of ecological problems resulting from landscape changes. The main goals of this study are: (1) to analyze the landscape changes of ecosystems in the Bohai coastal zone from 2000 to 2015 and (2) to identify the main driving forces of these changes during the past 15 years.

## 2. Study Area

The Bohai Sea is a “C”-shaped nearly enclosed sea. The scope of the coastal zone is a buffer zone, which is based on the coastline, and extends to a certain range across land and sea. There is no uniform standard for the division of the coastal zone at present, therefore, different researchers employ different definitions of coastal zones [27]. From the perspective of geomorphology, the coastal belt is a tideland zone between low and high tide. The widely accepted view is that the coastal zone is the area where the land interacts with the ocean. The coastal zone in this study is defined as the geographical area within the mainland with a distance of 10 km to the coastline, which is proposed as the basic unit in coastal zone evaluation. The coastal zone of the Bohai region spans three provinces and one municipality: Liaoning, Hebei, Shandong, and Tianjin, including 13 coastal cities: Dalian, Yingkou, Panjin, Jinzhou, Huludao, Qinhuangdao, Tangshan, Tianjin, Cangzhou, Binzhou, Dongying, Weifang, and Yantai from north to south, respectively (Figure 1).

Bohai bay plays an important role in national economic development. Administrations have proposed the conception of “Bohai Economic Rim” and distinctive development planning is implemented to boost coastal economy of Bohai bay. Driven by national policy, coastal landscapes are always under high development intensity and are being transformed at an unprecedent high rate in Bohai Bay. Coastal wetlands were reclaimed to meet the land needs for the construction of coastal ports, industrial parks, and coastal engineering projects. The construction scales of the Caofeidian industrial zone, Binhai new area of Tianjin, Huanghua port, and Dongying port have been gradually expanding since the implementation of the tenth “five-year plan,” inevitably changing the coastal ecosystem structure and spatial configuration.

## 3. Data and Methods

### 3.1. Data

Based on geographic information system (GIS) (ArcGIS 10.3) (ESRI, Redlands, CA, USA) technology, we have processed Landsat Thematic Mapper (TM) remote-sensing images of four typical periods in 2000, 2005, 2010 and 2015. The main data sources are Landsat TM images with 30 m spatial resolution during June to October. The ecosystem classification data came from the project “Survey and Assessment of National Ecosystem Changes between 2000 and 2010,” supported by the Ministry of Environmental Protection (MEP) of China and Chinese Academy of Sciences (CAS) [28], and the project “Survey and Assessment of National Ecosystem Changes between 2010 and 2015.” The research region is divided into seven categories: forest ecosystem (FE), shrub ecosystem (SE), grassland ecosystem (GL), wetland ecosystem (WE), farmland ecosystem (FL), urban ecosystem (UE), and bare land (BL). These ecosystems were subdivided into 24 classes.

### 3.2. Methods

#### 3.2.1. Landscape Changes

Changes in area, landscape dynamic degree, landscape change index, and the transition matrix of landscape types are four main indicators to depict landscape dynamics.

To quantitatively depict the range and speed of the ecosystems, the dynamic degree (*K*) was adopted and calculated. *K* refers to the percentage of ecosystem area changes per year within the initial ecosystem area, which can quantitatively reveal the ecosystem area change rate. The equation to calculate *K* is as follows [25,29]:(1)$$K=\;\frac{A_{t+1}-A_{t}}{A_{t}}\times \frac{1}{∆t}\times 100\%$$

*K* refers to the dynamic degree of land use for a specific ecosystem, defined as the percent of land use change per year, and *A_t_* and *A*_*t*+1_ represent the area of the coastal landscape for times *t* and *t*+1, ∆t means the duration of a certain period.

The landscape change index (LCI) is a good index to depict the overall landscape changes. The definition of the LCI is the absolute values of change in ecosystem types that have greatest impact on the formation of the landscape [30]. The LCI was calculated for each time interval by multiplying a factor of one-half by the sum of the absolute values of change in area proportion of each ecosystem type in relation to the total analyzed area, a constant of one-half was adopted to reflect the actual change level because summing the absolute values of change of each ecosystem type essentially doubled the index. The Equation for calculating LCI is:(2)$$LCI_{ti}=\frac{1}{2}\times \overset{n}{\underset{i=1}{\sum }}\left|CA_{i}\right|$$
where *LCI_ti_* represents the landscape change index in each time interval; $\left|CA_{i}\right|$ represents the absolute value of change in area proportion of each ecosystem type in relation to the total analyzed area, it was calculated with the following equation:(3)$$CA_{i}=(S_{t+1}/S_{t})/TA$$
where *CA_i_* represents the changes in area proportion of each ecosystem type in relation to the total area of research (%), *S*_*t*+1_ and *S_t_* represents the area of each ecosystem type during the time interval *t*+1 (km^2^) and *t* (km^2^); *TA* represents the total research area (km^2^).

The cross-tabulation matrix method [31] was used to analyze the landscape changes in three-time intervals (2000–2005, 2005–2010, 2010–2015). Table 1 showed the format of transition matrix. The columns display the categories of time 1 and the rows display the categories of time 2; *P_ij_* represents the proportion of the landscape that experiences a transition from category *i* to *j*; *P_ii_* represents the proportion of the landscape that shows persistence of category *i*; *P*_*i*+_ and *P*_+*j*_ represent the proportion of the landscape in category *i* in time 2 and category *j* in time 1 respectively. The column of gain and net change indicate the proportion of landscape that experiences gross gain and net change of each landscape type between time 1 and 2.

#### 3.2.2. Landscape Metrics

The spatial configuration of coastal landscapes is as much a reflection of the past as it is an indicator of the current socioeconomic processes and interactions [32]. Commonly, landscape metrics can be used to conduct empirical analysis of landscape pattern changes. A wide variety of metrics for characterizing landscapes have been proposed [33]. In view of numerous landscape metrics, we chose appropriate metrics based on four criteria: (1) comparability with previous research on landscape pattern changes; (2) ability to indicate ecological conditions of ecosystems; (3) low redundancy among landscape indices; and 4) ability to reflect the landscape pattern characteristics within the study area [34]. Based on the objectives of this paper and the general situation of the study area, four metrics at class level (NP, PD, MPS, and LPI) and five metrics at landscape level (NP, PD, MPS, LPI, and SHDI) were chosen to illustrate landscape pattern changes. Fragstats 4.2.1, developed by the Forest Science Department, Orgen State University, USA, is a program for quantifying landscape metrics for each period and analyzing the ecosystem spatial changes of the Bohai coastal region [35]. The definition and description of the landscape metrics are given in the Fragstats user’s guide [35]. The formulas [35] are as follows:
NP: Number of patchesNP = *n_i_*;*n*: the number of patches, NP ≥ 1, without limit.PD: Patch densityPD = *N*/*A*;*N*: number of patches; *A*: total landscape area, PD > 0, without limit.MPS: Mean patch sizeMPS = *A*/*N*;*N*: number of patches; *A*: total landscape area, MPS > 0, without limit.LPI: Largest patch indexLPI = Max(*a*_1_, …*a_n_*)/*A*×100;*a_i_*: area of patch *i*; *A*: total landscape area, 0 < LPI ≤ 100.SHDI: Shannon–Weaver diversity index$\mathrm{SHDI}=-\overset{m}{\underset{i\;=\;1}{\sum }}\left[P_{i}\ln\left(P_{i}\right)\right]$;*P_i_*: the proportion of landscape occupied by patch type *i*; *m*: number of patch types present in the landscape. SHDI ≥ 0.

#### 3.2.3. Driving Forces Analysis

Ordination is a widely used method which attempts to reveal the relationships between ecological landscapes and environmental variables [36]. Detrended correspondence analysis (DCA) was firstly performed to test the length of environmental gradients of the axes. The length of environmental gradient is 0.67, thus redundancy analysis (RDA) was used to analyze the relationships between landscape changes and environmental variables.

The relative socio-economic data in this study mostly came from the public statistical yearbook of history (2015). The driving force of ecosystem pattern change in the Bohai coastal zone includes seven factors: annual mean precipitation (APm, mm), annual mean temperature (ATm, ℃), population (Pop, person), GDP per capita (GDPpc, yuan/per capita), crop acreage (CA, 10^3^ hm^2^), grain yield (GY, 10^4^ tons), and output of aquatic products (AQ, 10^4^ tons). In consideration of different dimensions of driving factors, the deviation normalization method was adopted to preprocess the data. Then DCA and RDA were performed with Canoco 4.5 (Microcomputer Power, Ithaca, NY, USA) for Windows.

## 4. Results

### 4.1. Characteristics of Ecosystem Spatial Changes

#### 4.1.1. Spatial Distribution Characteristics of Ecosystems

Many ecosystems constitute the complex landscape pattern of the Bohai coastal zone. The total area and proportions of different ecosystems were shown in Table 2 and Figure 2 and Figure 3.

From 2000 to 2015, the wetland ecosystem had the largest area, accounting for about 39% of the total area, followed by farmland, urban ecosystem, and forest ecosystem. The area of these four ecosystems accounts for 90% of the total coastal zone (Figure 3).

Reservoir/pond was the main landscape type of the wetland ecosystem, accounting for 84.53% of the total wetland area; dry land was the predominant landscape type of the farmland ecosystem, accounting for 90.44% of the total farmland area; residential land accounted for 92.02% of the total urban area and was the dominant urban ecosystem type; broad-leaved forest has the largest area, accounting for 86.64% of the total forest area (Table 2).

#### 4.1.2. Characteristics of Ecosystem Spatial Changes

The statistics show that the areas of the observed ecosystems have all changed from 2000 to 2015 (Figure 4). Significant changes mainly occurred in urban, farmland, wetland and bare land ecosystems. The trend of landscape changes is characterized by the expansion of urban and the shrinkage of farmland, wetland and bare land. The urban ecosystems exhibited the largest area increment in these three time intervals. The largest decrease occurred in farmland ecosystems during 2000–2005 and 2005–2010. The area of wetland ecosystems showed the largest decrease during 2010–2015. The area of bare land continuously decreased from 2000–2015.

The dynamic degree (*K*) of all ecosystem types in the study area were calculated using Equation (1) (Table 3). During 2000–2005, the *K* value of grassland ecosystems was the largest, accounting for −8.46% of the change in total landscape, which indicated that the change amplitude of grassland was the biggest, followed by those of urban and bare land, accounting for 3.3% and −2.4% of the change in total landscape, respectively. From 2005 to 2010, the *K* value of urban ecosystems was the highest, accounting for 2.56% of the change in total landscape, followed by bare land and grassland, accounting for −1.59% and −1.51% of the change in total landscape. In the period 2010–2015, the *K* value of grassland ecosystems was the largest, accounting for 7.93% of the change in total landscape, followed by urban and bare land; the change of other ecosystems was comparatively small.

The landscape change index level (LCI) in the period 2000–2005 and 2010–2015 was relatively higher than the index for the period 2005–2010 (Table 3), indicating that the largest changes of the landscape occurred in the first and the third time interval.

In the period 2000–2005, farmland (FL) showed the largest loss, 2.24% of the entire coastal zone, and urban (UE) showed the largest gain, 2.64% of the coastal zone (Table 4). Most of the lost farmland converted into urban (UE). From 2000–2005, the largest net loss is farmland (FL) (−1.4%), and the largest net gain is urban (UE) (2.09%). The main contributor for the increment of urban was farmland (1.28%) and wetland (0.71%).

In the period of 2005–2010, wetland (WE) showed the largest loss, accounting for 3.23% of the entire coastal zone, and urban (UE) showed the largest gain, 3.46% of the coastal zone (Table 5). Most of the lost wetland converted to urban and farmland or degraded to bare land (BL). During this time interval, the largest net loss is FL (0.91%), and the largest net gain is urban (1.87%). The largest contributor for the increment of urban was farmland and wetland.

During 2010–2015, the largest loss occurred in wetland ecosystems (WE), accounting for 3.70% of the total coastal zone, and urban (UE) showed the largest gain, 2.38% of the total coastal zone (Table 6). Most of the lost wetland was converted to farmland or degraded to bare land. The largest net loss is wetland (1.98%) and the largest net gain is urban (2.06%). Wetland was the largest contributor for the increment of the urban area.

### 4.2. Quantification of Landscape Metrics

The quantification of landscape pattern through landscape metrics is an effective way to analyze landscape pattern changes [33,37]. Table 7 compares changes in the landscape metrics at the class level in 2000, 2005, 2010 and 2015. Grasslands were the only decreasing landscape patch type: NP decreased from 606 in 2000 to 494 in 2015. The wetland NP increased the most, from 2070 in 2000 to 3053 in 2015, while MPS decreased from 282.35 km^2^ to 180.33 km^2^, which indicated that the wetland ecosystem had become more fragmented. The NP and MPS of forest, shrub, farmland, and bare land ecosystems had the same changing trend as the wetlands. Therefore, the fragmentation of these ecosystems increased. Both urban NP and MPS exhibited an increasing trend. The LPI for the wetland was the largest in 2000, 2005, 2010 and 2015, indicating that the wetland ecosystem was the dominant landscape type in the Bohai coastal zone. The results revealed that spatial patterns have changed significantly; the change direction of these ecosystems has been toward increased heterogeneity and fragmentation.

A comparison of the landscape indices at the landscape level is listed in Table 8. NP increased from 14,663 in 2000 to 17,732 in 2015, while MPS steadily decreased to 83.61 km^2^ from 101.14 km^2^, indicating that some original patches were divided, and landscape heterogeneity and fragmentation were rising. SHDI has not changed much.

### 4.3. Driving Forces

Preliminary DCA estimated a gradient length of 0.67 SD, and thus, the use of RDA as a linear method of canonical ordination was appropriate. The results of RDA were shown in a bi-plot (Figure 5). The full RDA model using seven environmental variables explained 80.3% of the total variance within the ecosystem landscape changes (Table 9). RDA results indicated varying correlations between each ecosystem type and environmental variable. AQ and Pop were the main factors related to the change of the wetland ecosystem. Pop and GDPpc were closely related to the change of the urban ecosystem. ATm, CA, and GY had positive correlations with the change of the farmland ecosystem.

## 5. Discussion

### 5.1. Land Reclamation in Bohai Coastal Zone

Bohai coastal areas are one of the most densely populated regions in China because of their prominent biological productivity and high accessibility. Spurred by a fast-growing economy and large population, coastal administrations have aggressively expanded seafaring facilities and constructed coastal industries to promote urbanization, which has resulted in a great demand for land area [38]. As a feasible land solution for coastal development, land reclamation can alleviate the pressure of land shortages. The Changlu sea salt pan, one of China’s four salt fields, is located along northern Bohai Bay. Driven by huge profit, the total area of the Changlu sea salt pan has been expanded from 216 km^2^ to 6580 km^2^ from 1949 to 1965 [38]. In the 21st century, the Bohai coastal region has become a focus of economic development with the promotion of national policies in the Bohai Economic Rim. Since 2000, some of the reclaimed areas have been used for mariculture and agriculture, but most of them are used to accommodate the demand for transport, employment and other urban facilities. The Tangshan Caofeidian Industrial Zone and Tianjin Binhai New area are typical examples of coastal reclamation projects. Caofeidian Industrial District is located in the heartland of Bohai Bay. The reclamation plan was carried out in 2004 and is set to be finished in 2020. About 310 km^2^ waterfront area was reclaimed for deep water port and for steel, chemical, electric works, and nuclear power industries [38]. Tianjin New District is another example, nearly 2270 km^2^ of reclaimed area was used for the construction of new housing, ports and other coastal industries [39]. Furthermore, some coastal provinces and metropolises have developed their own coastal economy development plans to boost the marine economy. For instance, in Shandong province, the “Special Plan on Focused and Intensive Sea Use for Blue Economic Zone Construction around Shandong Peninsula (2009–2020)” proposed implementing focused and intensive use of the sea by setting up a total of nine big and 10 small costal industrial complexes, covering a total area of 1500 km^2^, including 520 km^2^ from reclamation [40]. Reclamation has greatly altered the spatial distribution of ecosystems of the Bohai coastal zone, wetlands have been transformed into farmland and urban or degraded land into bare land. Over the last 20 years, insufficient ecological considerations and inadequate coastal protection measures have resulted in sharp area reduction and ecosystem service declines of wetland ecosystems. Presently, the Chinese government has recognized the conflicts between wetland ecosystem protection and the space demands from a fast-growing economy. The Chinese government has committed to following the concept of sustainable development via policy, legislation, science, and management approaches, by strengthening laws and regulations and improving coastal spatial planning so that ecological protection can be properly taken into management consideration [38]. These efforts lay a good foundation for coastal restoration and scientific decision-making or management of coastal conservation.

### 5.2. Landscape Changes and Their Impact on Coastal Habitat

The above results indicated that the coastal landscape pattern in Bohai has changed dramatically over the past 15 years. The natural wetland consistently decreased from 2000 to 2015 and became fragmentated and heterogenous, with a significant decrease in marshland and lake areas. Thus, the habitat gradually shrunk and diminished. Waterbirds are crucial indicators for assessment of ecosystem health [41,42]. The species and number of waterbirds can reflect the health status of wetland ecosystem. Extensive mudflats of northern Bohai Bay have formerly supported over 65,000 Red Knots, 60% of the entire flyway population, and 80,000 Curlew Sandpipers (45% of the population) on their northward migration [43]. However, in recent years, approximately 450 km^2^ of offshore area, including 218 km^2^ of intertidal flats have been reclaimed for the Caofeidian Industrial District and Tianjin Industrial District, which resulted in increasing loss and degradation of waterbird habitat. Only a small stretch of mudflats remains in this area, forcing the northward migrating birds to concentrate and crowd into the small remaining area [44]. It was reported that the spring peak numbers of Curlew Sandpiper *C. ferruginea* increased from 3% in 2007 to 23% in 2010 of the flyway population [44]. Fragmentation and loss of waterbird habitat leads to a decline in waterbird numbers [45,46] or to the movement of birds to nearby suitable habitats [47]; the latter can lead to increased densities at other sites and consequently an increase in mortality of the displaced birds, leading to an overall loss of birds [48]. Landscape modification and habitat fragmentation have negative impacts on coastal ecosystems and biodiversity because Bohai Bay is a crucial area for waterbirds to make stopovers or winter here. It is important that decision makers and the public are made fully aware of the great importance of Bohai Bay for waterbirds and of the potential environmental disasters for all those that rely on the tidal flats as a food source [49].

### 5.3. Limitatios of Driving Forces Analysis

By selecting two natural and five socio-economic factors, we examined the relationship between landscape changes and driving forces. However, the study area in this research is a buffer zone, it is very difficult to gain corresponding relative socio-economic data. We used socio-economic data of coastal cities to conduct the analysis of driving forces. A bivariate correlation method was employed to testify to the feasibility of driving forces in the case of inconsistent analysis units. We used spatial distribution data on population (POPs) and GDP (GDPs) and calculated mean values of POPs and GDPs with ArcGIS 10.3, named POPsm and GDPsm. Then the relationships between POPsm and Pop, GDPsm and GDP were analyzed in SPSS 19.0 (IBM, Armonk NY, USA). The results indicated that Pop showed significant correlation with POPsm, and GDP showed extremely significant correlation with GDPsm (Table 10). The results of correlation analysis indicated that socio-economic data of coastal cities can be used to conduct the analysis of driving forces.

In addition, Due to unavailable data, unknown influencing factors and factors that are impossible to quantify, it is impossible to include all socio-economic and environmental variables that influence landscape changes [50]. Natural factors, such as elevation, slope direction and altitude are influential in reshaping landscapes. Whereas further elaborate analysis wasn’t performed due to data limitation. In this study, we merely analyzed and displayed the consequences of landscape spatial changes and paid less attention to the landscape process and function. The difference between correlation and causality should be distinguished in future researches of landscape changes.

The correlation analysis is helpful for the identification of influential factors, the magnitude of factors is not quantified yet. Focusing on the changes in landscape processes and functions is helpful for the identification of dominant factors, and then it is possible to further explore the driving mechanism of landscape changes.

## 6. Conclusions

The ecosystem landscape pattern in the Bohai coastal zone has changed significantly over the 15 years of this research. The wetland ecosystem maintained the largest area, followed by the farmland and urban ecosystems. The main trend of landscape changes was characterized by the expansion of the urban area and the shrinkage of farmland, wetland and bare land.

There was no obvious changing trend in dynamic degree (*K*). The analysis of dynamic degree showed that the extent of grassland change was the largest from 2000–2005 and 2010–2015, accounting for −8.46% and 7.93% of the change in landscape, respectively. The *K* value of urban was the highest from 2005–2010, accounting for 2.56% of the change in total landscape. The landscape change index level (LCI) in the period 2000–2005 and 2010–2015 was relatively higher than the index for the period 2005–2010, indicating that the largest changes of the landscape occurred in the first and the third time interval.

The transformation of landscape types mainly occurred in wetland, farmland, urban and bare land ecosystems. The farmland showed the largest loss and urban showed the largest gain during 2000–2005, and farmland was the main contributor for the increment of urban. During 2005–2010 and 2010–2015, the wetland showed the largest loss and urban still showed the largest gain; most of the wetland converted to farmland and urban or degraded to bare land; wetland and farmland were the main contributors for the expansion of urban.

The overall NP increased, whereas MPS decreased, indicating that the spatial landscape pattern had become fragmented and heterogeneous.

The results of the RDA model show that landscape changes are indicated by socio-economic and natural variables. Human activity is a major driving force in shaping the spatial distribution of the ecosystems.

## Figures and Tables

**Figure 1 ijerph-16-00536-f001:**
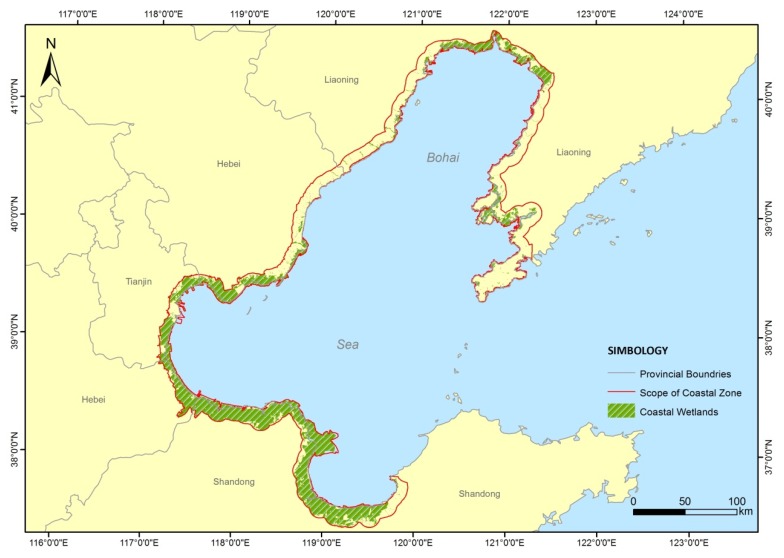
Location of the study area in China.

**Figure 2 ijerph-16-00536-f002:**
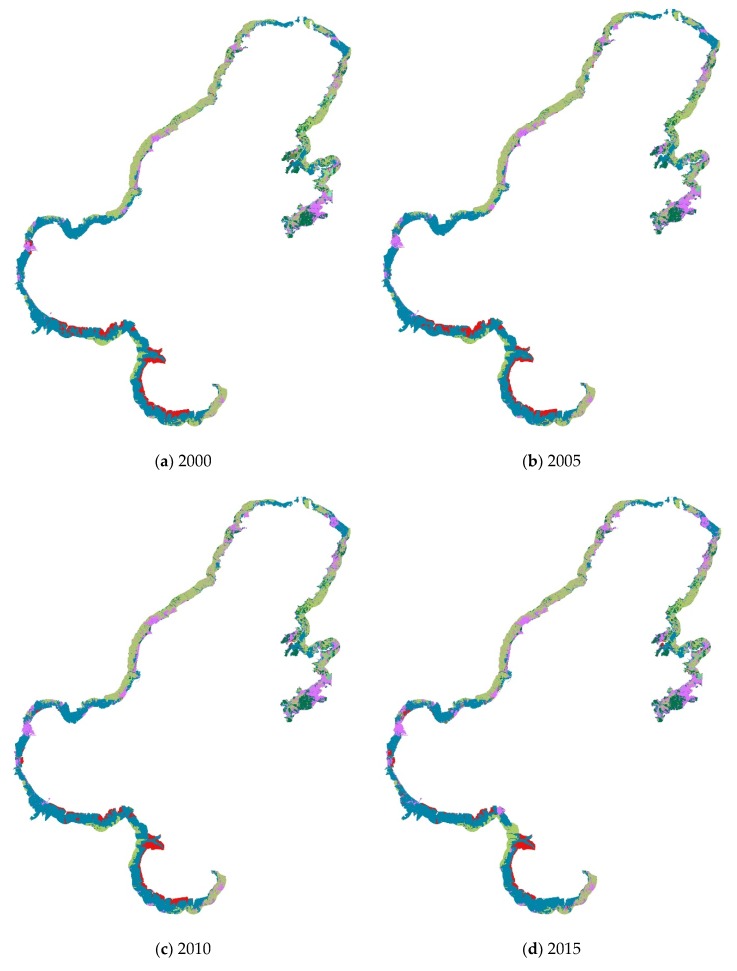
Landscape changes of Bohai coastal zone between 2000 and 2015: (**a**) map of ecosystems in 2000; (**b**) map of ecosystems in 2005; (**c**) map of ecosystems in 2010; (**d**) map of ecosystems in 2015.

**Figure 3 ijerph-16-00536-f003:**
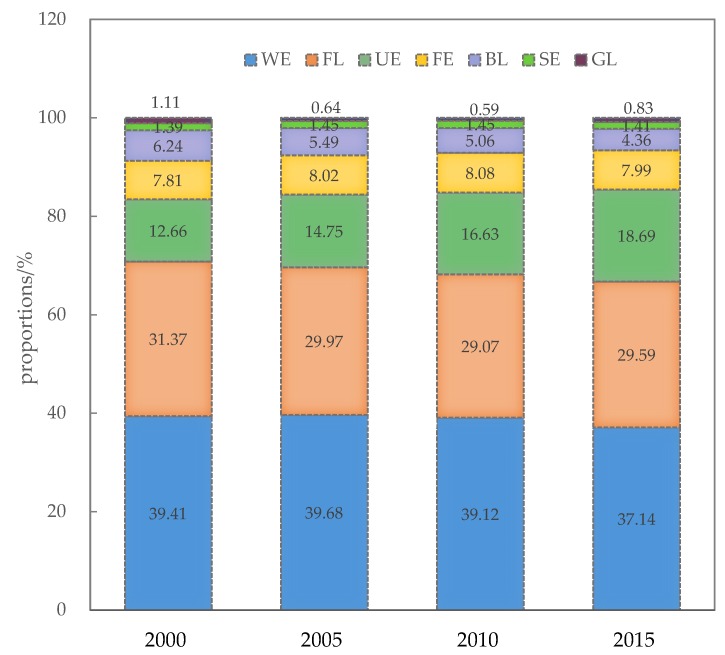
Proportions of different ecosystems in 2000, 2005, 2010 and 2015.

**Figure 4 ijerph-16-00536-f004:**
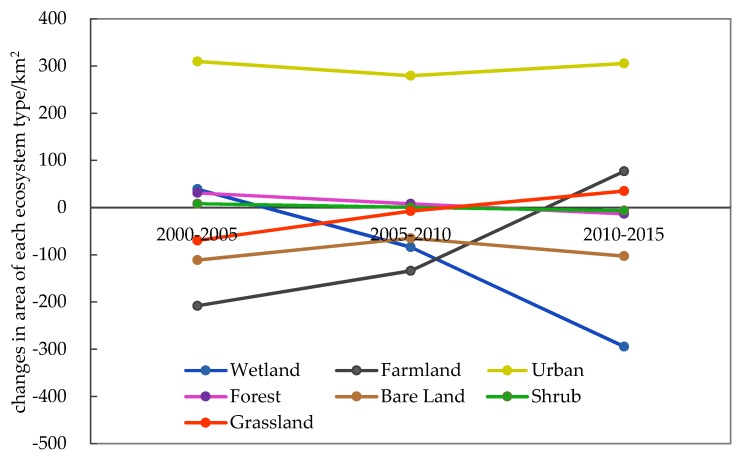
Area changes of ecosystems from 2000 to 2015.

**Figure 5 ijerph-16-00536-f005:**
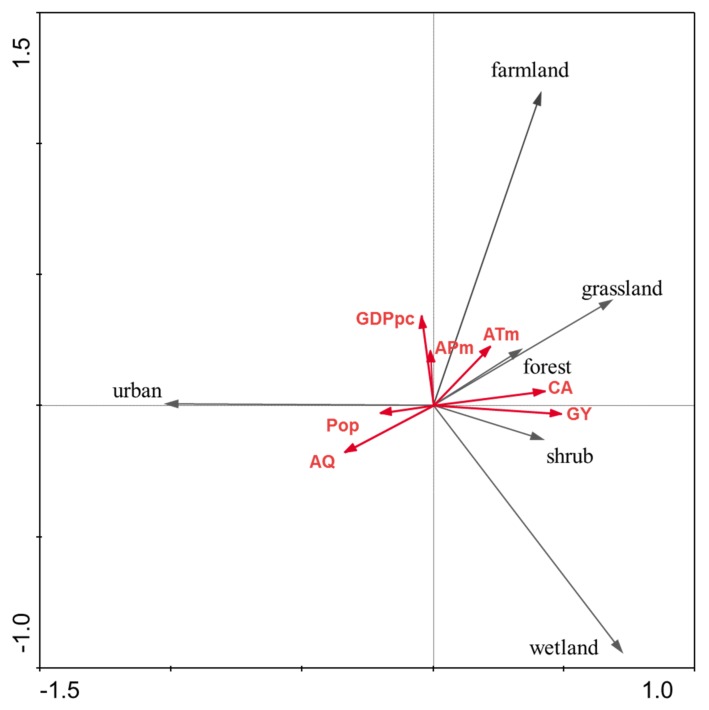
Redundancy analysis (RDA) of ecosystem landscape changes in relation to combined socio-economic and environmental variables from 2000 to 2015.

**Table 1 ijerph-16-00536-t001:** Transition matrix for comparing landscape changes in time [31].

	Time 1	Category 1	Category 2	Category 3	Category 4	Total Time 2	Gain	Net Change
Time 2								
Category 1		*P* _11_	*P* _21_	*P* _31_	*P* _41_	*P* _+1_	*P*_+1_ − *P*_11_	*P*_+1_ − *P*_1+_
Category 2		*P* _12_	*P* _22_	*P* _32_	*P* _42_	*P* _+2_	*P*_+2_ − *P*_22_	*P*_+2_ − *P*_2+_
Category 3		*P* _13_	*P* _23_	*P* _33_	*P* _43_	*P* _+3_	*P*_+3_ − *P*_33_	*P*_+3_ − *P*_3+_
Category 4		*P* _14_	*P* _24_	*P* _34_	*P* _44_	*P* _+4_	*P*_+4_ − *P_44_*	*P*_+4_ − *P*_4+_
Total time 1		*P* _1+_	*P* _2+_	*P* _3+_	*P* _4+_	1		
Loss		*P*_1+_ − *P*_11_	*P*_2+_ − *P*_22_	*P*_3+_ − *P*_33_	*P*_4+_ − *P*_44_			

**Table 2 ijerph-16-00536-t002:** Area and proportion of ecosystems from 2000 to 2015.

Ecosystem Types		2000		2005		2010		2015	
		Area (km^2^)	Proportion (%)	Area (km^2^)	Proportion (%)	Area (km^2^)	Proportion (%)	Area (km^2^)	Proportion (%)
WE ^1^	Marsh land	1113.5	7.5	1022.0	6.9	804.1	5.4	599.6	4.0
	Lake	7.0	0.0	8.1	0.1	5.0	0.0	5.1	0.0
	Reservoir/Pond	4515.9	30.5	4640.3	31.3	4757.7	32.1	4653.8	31.4
	River	208.2	1.4	213.4	1.4	233.5	1.6	247.2	1.7
FL ^2^	Paddy field	409.6	2.8	348.6	2.4	363.2	2.4	378.1	2.6
	Dry land	4206.1	28.4	4054.3	27.3	3904.3	26.3	3966.9	26.8
	Garden plot	36.1	0.2	40.7	0.3	41.9	0.3	41.4	0.3
UE ^3^	Residential land	1520.0	10.3	1717.8	11.6	2270.0	15.3	2549.9	17.2
	Urban green land	75.6	0.5	78.7	0.5	82.1	0.6	88.7	0.6
	Transportation land	73.7	0.5	89.1	0.6	104.3	0.7	118.6	0.8
	Mining area	9.0	0.1	10.4	0.1	9.3	0.1	13.8	0.1
	Industrial land	198.5	1.3	290.5	2.0	0.0	0.0	0.0	0.0
FE ^4^	Broad-leaved forest	1006.5	6.8	1029.1	6.9	1038.9	7.0	1026.0	6.9
	Coniferous forest	66.2	0.4	71.6	0.5	68.9	0.5	70.3	0.5
	Mixed broadleaf-conifer forest	85.9	0.6	89.0	0.6	89.6	0.6	87.4	0.6
	Sparse forest	0.0	0.0	0.0	0.0	0.0	0.0	0.5	0.0
BL ^5^	Bare land	925.7	6.2	814.4	5.5	749.5	5.1	646.4	4.4
SE ^6^	Broadleaf shrub	206.4	1.4	214.5	1.4	215.3	1.5	208.2	1.4
	Acerola shrub	0.1	0.0	0.1	0.0	0.2	0.0	0.2	0.0
	Sparse shrub	0.0	0.0	0.0	0.0	0.0	0.0	0.5	0.0
GL ^7^	Meadow	0.0	0.0	0.0	0.0	1.5	0.0	0.2	0.0
	Prairie	77.2	0.5	7.9	0.1	0.0	0.0	0.6	0.0
	Tussock	44.3	0.3	58.4	0.4	55.2	0.4	49.0	0.3
	Sparse grassland	43.6	0.3	29.0	0.2	31.4	0.2	73.2	0.5

^1^ Wetland ecosystems; ^2^ farmland ecosystems; ^3^ urban ecosystems; ^4^ forest ecosystems; ^5^ bare land; ^6^ shrub ecosystems; ^7^ grassland ecosystems.

**Table 3 ijerph-16-00536-t003:** Dynamics of landscape changes in ecosystems in the research area during 2000–2015.

Time Interval	Indicator	Ecosystem Types						
		WE ^1^	FL ^2^	UE ^3^	FE ^4^	BL ^5^	SE ^6^	GL ^7^
2000–2005	*K*/%	+0.13	−0.89	+3.30	+0.54	−2.40	+0.79	−8.46
	CA/%	0.26	−1.40	2.09	0.21	−0.75	0.05	−0.47
	LCI	2.62						
2005–2010	*K*/%	−0.28	−0.60	+2.56	+0.13	−1.59	0.08	−1.51
	CA/%	−0.56	−0.91	1.88	0.05	−0.44	0.01	−0.05
	LCI	1.95						
2010–2015	*K*/%	−1.02	+0.36	+2.48	−0.22	−2.75	−0.61	+7.93
	CA/%	−1.99	0.52	2.06	−0.09	−0.69	−0.04	0.24
	LCI	2.81						

^1^ Wetland ecosystems; ^2^ farmland ecosystems; ^3^ urban ecosystems; ^4^ forest ecosystems; ^5^ bare land; ^6^ shrub ecosystems; ^7^ grassland ecosystems; “+*K*” indicates the increase; “−*K*” indicates the decrease.

**Table 4 ijerph-16-00536-t004:** Landscape transition matrix between 2000 (columns) and 2005 (rows).

	2000	GL	UE	SE	BL	FL	FE	WE	Total	Gain	Net Change
2005											
GL		0.41	0.01	0.00	0.01	0.05	0.00	0.17	0.64	0.23	−0.47
UE		0.30	12.10	0.00	0.15	1.28	0.20	0.71	14.74	2.64	2.09
SE		0.00	0.00	1.38	0.00	0.04	0.01	0.01	1.45	0.07	0.05
BL		0.02	0.01	0.00	4.69	0.03	0.00	0.73	5.49	0.80	−0.75
FL		0.08	0.38	0.01	0.03	29.14	0.09	0.26	29.98	0.84	−1.40
FE		0.17	0.02	0.00	0.01	0.27	7.49	0.05	8.02	0.53	0.21
WE		0.13	0.13	0.00	1.35	0.57	0.02	37.47	39.68	2.20	0.27
Total		1.11	12.65	1.39	6.24	31.38	7.81	39.41	100.00		
Loss		0.70	0.55	0.02	1.55	2.24	0.32	1.94			

**Table 5 ijerph-16-00536-t005:** Landscape transition matrix between 2005 (columns) and 2010 (rows).

	2005	GL	UE	SE	BL	FL	FE	WE	Total	Gain	Net Change
2010											
GL		0.38	0.02	0.00	0.01	0.02	0.00	0.14	0.58	0.20	−0.43
UE		0.05	13.14	0.07	0.07	1.81	0.25	1.21	16.61	3.46	1.87
SE		0.00	0.07	1.21	0.00	0.09	0.07	0.01	1.45	0.24	0.01
BL		0.02	0.01	0.00	3.52	0.03	0.02	1.40	5.01	1.49	−0.39
FL		0.05	0.97	0.08	0.06	27.14	0.40	0.42	29.12	1.98	−0.91
FE		0.01	0.24	0.07	0.01	0.47	7.24	0.05	8.09	0.84	0.05
WE		0.12	0.28	0.01	1.72	0.47	0.05	36.48	39.14	2.65	−0.57
Total		0.64	14.74	1.45	5.40	30.03	8.03	39.71	100.00		
Loss		0.26	1.60	0.24	1.88	2.88	0.79	3.23			

**Table 6 ijerph-16-00536-t006:** Landscape transition matrix between 2010 (columns) and 2015 (rows).

	2010	GL	UE	SE	BL	FL	FE	WE	Total	Gain	Net Change
2015											
GL		0.52	0.02	0.00	0.00	0.00	0.00	0.28	0.83	0.31	0.24
UE		0.04	16.31	0.04	0.05	0.81	0.18	1.25	18.69	2.38	2.06
SE		0.00	0.01	1.39	0.00	0.01	0.01	0.00	1.41	0.02	−0.04
BL		0.01	0.01	0.00	3.66	0.01	0.01	0.65	4.36	0.70	−0.70
FL		0.00	0.15	0.01	0.02	27.86	0.05	1.51	29.59	1.74	0.52
FE		0.00	0.04	0.01	0.00	0.13	7.80	0.02	7.99	0.19	−0.09
WE		0.02	0.09	0.00	1.32	0.26	0.03	35.42	37.14	1.72	−1.98
Total		0.59	16.63	1.45	5.05	29.07	8.08	39.12	100.00		
Loss		0.07	0.32	0.07	1.39	1.22	0.27	3.70			

**Table 7 ijerph-16-00536-t007:** Landscape pattern metrics of ecosystems at the class level.

Metrics at Class Level		Wetland	Farmland	Urban	Forest	Bare Land	Shrub	Grassland
NP	2000	2070	2230	5974	2016	750	1017	606
	2005	2267	2478	6019	1002	2198	702	536
	2010	2676	2760	6436	2457	788	1058	461
	2015	3053	3028	6619	2534	919	1085	494
PD	2000	0.14	0.15	0.40	0.14	0.05	0.07	0.04
	2005	0.15	0.17	0.41	0.07	0.15	0.05	0.04
	2010	0.18	0.19	0.43	0.17	0.05	0.07	0.03
	2015	0.21	0.20	0.45	0.17	0.06	0.07	0.03
MPS	2000	282.35	208.56	31.43	57.48	123.43	20.32	27.26
	2005	259.55	179.29	36.34	21.43	54.14	116.01	17.80
	2010	216.74	156.14	38.31	48.73	95.11	20.37	19.13
	2015	180.33	144.87	41.87	46.72	70.31	19.25	24.92
LPI	2000	18.86	6.06	2.33	1.52	0.93	0.16	0.24
	2005	14.13	4.01	2.61	0.16	1.49	1.26	0.07
	2010	13.84	2.85	2.38	1.50	1.56	0.16	0.07
	2015	10.24	2.64	2.63	1.49	1.55	0.16	0.08

NP: number of patches; PD: patch density; MPS: mean patch size; LPI: largest patch index.

**Table 8 ijerph-16-00536-t008:** Landscape pattern metrics of ecosystems at the landscape level.

Landscape Level	NP	PD	MPS	LPI	SHDI
2000	14,663	0.99	101.14	18.86	1.47
2005	15,202	1.03	97.54	14.13	1.47
2010	16,636	1.12	89.12	13.84	1.47
2015	17,732	1.20	83.61	10.24	1.48

NP: number of patches; PD: patch density; MPS: mean patch size; LPI: largest patch index; SHDI: Shannon-Weaver diversity index.

**Table 9 ijerph-16-00536-t009:** Summary statistics for first four axes of RDA with the landscape changes and environmental variables for ecosystems in the Bohai coastal zone.

Axes	Axis 1	Axis 2	Axis 3	Axis 4	Total Variance
Eigenvalues	0.384	0.227	0.078	0.071	1.000
Species-environment correlations	0.984	0.946	0.880	0.714	
CV of species data	38.4	61.1	68.9	76.0	
CV of species-environment relation	47.8	76.1	85.8	98.0	
Sum of all eigenvalues					1.000
Sum of all canonical eigenvalues					0.803

CV: cumulative percentage variance.

**Table 10 ijerph-16-00536-t010:** The relationship analysis between Pop and POPsm as well as GDP and GDPsm.

Items		Pop	Items		GDP
POPsm	Pearson correlation	0.627 *	GDPsm	Pearson correlation	0.886 **
	*p* value	0.022		*p* value	<0.01

* Coefficient is significant at the 0.05 level; ** Coefficient is significant at the 0.01 level; POPsm: mean values of POPs; GDPsm: mean values of GDPs.

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
