# Peer review of "Ecosystem Spatial Changes and Driving Forces in the Bohai Coastal Zone"

_ijerph, 2019, doi:10.3390/ijerph16040536_

Reviewer 1 Report

Review of manuscript entitled: Ecosystem spatial changes and driving forces in the Bohai coastal zone       

Analysis and understanding of landscape changes is a key issue in the context of planning landscape in the future. From this point of view it is important to obtain high accuracy land cover data. Remote sensing images used by authors allow to obtain such accuracy. This in turn allows for correct analysis of landscape evolution.The authors of the manuscript  made very detailed analyses of landscape transformations based mainly on analyses of indicators showing landscape structur and intensity of changes for each ecosystem separately. Despite the analyses are very detailed and show many landscape features it would be worth to supplement them with eg. analysis of landscape change index (LCI).

You can find proper reference here:

https://www.mdpi.com/2071-1050/10/12/4526

https://www.researchgate.net/publication/299854923_Landscape_changes_in_selected_suburban_area_of_Bratislava_Slovakia

You should at least mention in the discussion section that there are some indicators determining the level of landscape changes as a whole. This indicator would allow you to determine the level of landscape changes for the entire analyzed area in each period of time and compare it. Research generally stand at a high level but it required some corrections:

1) Introduction section is very short and should be improved by adding some more references to analyses of driving forces of landscape change in Europe.

2) Methods are clear and well described

3) Results should be improved - landscape change matrix is prepared for 3 time intervals (2000-2005, 2005-2010, 2010-2015), but other results show analyses, graphs for one period from 2000 to 2015. That's why I cannot see any trends in changes of landscape metrics. It is a little bit confused.4) Discussion section can be improved by adding information of limitations and possible errors/mistakes of the study as well as directions of further studies in this field.

5) Conclusions are supported by results.

Author Response

1)     Introduction section is very short and should be improved by adding some more references to analyses of driving forces of landscape change in Europe.

Response 1: We conduct literature research on driving forces of landscape change in Europe. The definition, classification of driving forces was supplemented in introduction section (Page 2, Line 17-32). Forces that cause observed landscape changes are labeled keystone processes or drivers[1]. Five major types of driving forces have been identified: socio-economic, political, technological, natural, and cultural driving forces[2]. Understanding of these driving forces may be helpful in reconstructing past landscape changes and in predicting future changes[3], and therefore help to elaborate sustainable management practices aimed at preserving essential ecosystem functions and services[4, 5, 6].

2)     Methods are clear and well described

Response 2: According to your suggestion, we added landscape change index (LCI) to determine the intensity level of landscape changes[7]. The description and calculation formula of LCI was supplemented in section data and methods (Page 4, Line 8-22). LCI was listed in table 2 (Page 9). The analysis of LCI was added in section 4.1.2 (Page 8, Line 27-29).

3) Results should be improved - landscape change matrix is prepared for 3 time intervals (2000-2005, 2005-2010, 2010-2015), but other results show analyses, graphs for one period from 2000 to 2015. That's why I cannot see any trends in changes of landscape metrics. It is a little bit confused.

Response 3: I’m really sorry for the confusion. We have examined the landscape change matrix for each time interval, but we merely listed landscape matrix from 2000 to 2015 in order to exhibit the overall landscape changes in the past 15 years. The cross-tabulation matrix method[8] was employed to identify the trends in landscape changes (Page 4, Line 23-31). The landscape transition matrixes from 2000-2005, 2005-2010, 2010-2015 were list in table 4 (Page 9), table 5 (Page 10) and table 6 (Page 10). Metrics and class and landscape level in 2005 and 2010 were added to table 7 (Page 11) and table 8(Page 11).

3)     Discussion section can be improved by adding information of limitations and possible errors/mistakes of the study as well as directions of further studies in this field. 

Response 4: Thank you very much for your valuable suggestions. Section 5.3, the limitations of driving forces (Page 14, Line 5-18) was added to improve the integrality of the discussion section. It is impossible to include all socio-economic data and environmental variables that influence landscape due to unavailable data, unknown influencing factors and factors that are impossible to quantified. Furthermore, most of landscape studies mainly focused on the documentation and analysis of spatial patterns and have paid less attention to landscape process and function. The study on driving mechanism of landscape process and function changes should be emphasized in future study.

5) Conclusions are supported by results.

 References:

1      Marcucci D J. Landscape history as a planning tool[J]. Landscape and Urban Planning. 2000, 49(1): 67-81.

2      Burgi M, Hersperger A M, Schneeberger N. Driving forces of landscape change - current and new directions[J]. Landscape Ecology. 2004, 19(8): 857-868.

3      Zhang Z, van Coillie F, Ou X, et al. Integration of Satellite Imagery, Topography and Human Disturbance Factors Based on Canonical Correspondence Analysis Ordination for Mountain Vegetation Mapping: A Case Study in Yunnan, China[J]. Remote Sensing. 2014, 6(2): 1026-1056.

4      Mladenoff D J. LANDIS and forest landscape models[J]. Ecological Modelling. 2004, 180(1): 7-19.

5      Syphard A D, Franklin J. Spatial aggregation effects on the simulation of landscape pattern and ecological processes in southern California plant communities[J]. Ecological Modelling. 2004, 180(1): 21-40.

6      Malanson G P, Wang Q, Kupfer J A. Ecological processes and spatial patterns before, during and after simulated deforestation[J]. Ecological Modelling. 2007, 202(3-4): 397-409.

7      Krajewski P, Solecka I, Mrozik K. Forest Landscape Change and Preliminary Study on Its Driving Forces in Ĺšlęża Landscape Park (Southwestern Poland) in 1883–2013[J]. Sustainability. 2018, 10(12): 4526.

8      Pontius R G, Shusas E, Mceachern M. Detecting important categorical land changes while accounting for persistence[J]. Agriculture, Ecosystems & Environment. 2004, 101(2-3): 251-268.

Reviewer 2 Report

Revision ijerph-418693

Title: Ecosystem Spatial Changes and Driving Forces in the Bohai Coastal Zone

APPRECIATION

This manuscript discusses the spatial changes of forest, shrub, grassland, wetland, farmland, urban and wetland ecosystems, as well as their landscape patterns, in the coastal area of Bohai (China), from 200 to 2015, and related these changes to driving forces (e.g., gross domestic product) using Redundancy Analysis.

The study is well conducted and written. The language is clear and the results / discussion are quite reasonable. However, the article lacks an adequate spatial representation of the land use / landscape changes. All the discussion is supported by tables and diagrams, but no maps are presented besides the location map. In the revised manuscript a thourough discussion of the spatial changes supported by maps is required.

RECOMMENDATION

Moderate revision

31 December 2018

Author Response

This manuscript discusses the spatial changes of forest, shrub, grassland, wetland, farmland, urban and wetland ecosystems, as well as their landscape patterns, in the coastal area of Bohai (China), from 200 to 2015, and related these changes to driving forces (e.g., gross domestic product) using Redundancy Analysis.

The study is well conducted and written. The language is clear and the results / discussion are quite reasonable. However, the article lacks an adequate spatial representation of the land use / landscape changes. All the discussion is supported by tables and diagrams, but no maps are presented besides the location map. In the revised manuscript a thorough discussion of the spatial changes supported by maps is required.

Response 1: Thank you very much for your encouragement and suggestion. The spatial distribution of landscape types in 2000, 2005, 2010 and 2015 had been supplemented in section 4.1.1 to represent the spatial changes (Figure 2) (Page 6). 

Reviewer 3 Report

This paper examined ecosystem spatial changes in the Bohai coastal zone using remote sensing (RS) images acquired in 2000, 2005, 2010, and 2015. The work calculated the landscape change and landscape metrics, and then explored their driving forces. All the methods are conventional, and the results do not offer readers with new understanding of the ecosystem spatial change in the Bohai coastal zone. Overall, I see no new methods and new results that make the work publishable in an international journal.

Author Response

This paper examined ecosystem spatial changes in the Bohai coastal zone using remote sensing (RS) images acquired in 2000, 2005, 2010, and 2015. The work calculated the landscape change and landscape metrics, and then explored their driving forces. All the methods are conventional, and the results do not offer readers with new understanding of the ecosystem spatial change in the Bohai coastal zone. Overall, I see no new methods and new results that make the work publishable in an international journal.

Response 1: Bohai coastal areas are one of the most densely populated regions in China, and coastal ecosystems provide easily accessible good and services to coastal communities. The sustainable development of coastal economy and the quality of human life closely depend on coastal landscapes and the crucial services they produced. Driven by national policy and large population, part of coastal ecosystems was reclaimed to promote urbanization. Thus, Coastal landscapes in Bohai bay have experienced high development intensity and unprecedent rate of transformation. The shrinkage and degradation of natural coastal ecosystems have negatively affected the quality and quantity of ecosystem services they provided. The analysis of past landscape changes and driving forces may be helpful to predict future changes and elaborate sustainable management practices aimed at preserving essential ecosystem functions and services. 

Reviewer 4 Report

This manuscript aims at examining the landscape changes of ecosystems in the Bohai coastal zone from 2000 to 2015 and identifying the main driving forces of these changes during the past 15 years. However, in my opinion, there are several obscure parts in the manuscript that prevent me to recommend publication as it is. The interest of the topic under analysis and the novelties of the empirical strategy lead me to encourage the authors to work on a serious major revision of the manuscript. The concerns that should be addressed in the revision are listed next. 1. The scope of the coastal zone is a buffer zone? How to gain corresponding relative socio-economic data? In my opinion, consistent analysis unit is a basic point of driving forces. If you used 13 coastal cities’ socio-economic data to replace socio-economic data in buffer zone is obviously wrong. 2. Why these seven factors were selected to investigate driving forces of landscape pattern changes? 3. There are many spelling and unambiguous problems. Such as, line 43 in page 1, …and and urban; line 18 in page 3, …of wetland area changes per year, why only used to wetland area changes; line 1 in page 4, periods: 2000–005? In table 1, what is meaning of Pp/%? In table2, the numbers should be reorganized, such as 1668.11, 1089.78. In line 6 in page 8, what is F? In line 24 in page 10, “Of”. “5.2. Impacts of the Reduction of Wetland Ecosystem on Birds” is not directly related to the purpose of paper.

Author Response

This manuscript aims at examining the landscape changes of ecosystems in the Bohai coastal zone from 2000 to 2015 and identifying the main driving forces of these changes during the past 15 years. However, in my opinion, there are several obscure parts in the manuscript that prevent me to recommend publication as it is. The interest of the topic under analysis and the novelties of the empirical strategy lead me to encourage the authors to work on a serious major revision of the manuscript. The concerns that should be addressed in the revision are listed next.

1.      The scope of the coastal zone is a buffer zone? How to gain corresponding relative socio-economic data? In my opinion, consistent analysis unit is a basic point of driving forces. If you used 13 coastal cities’ socio-economic data to replace socio-economic data in buffer zone is obviously wrong.

Response 1: The scope of the coastal zone is a buffer zone, and it is very difficult to gain corresponding relative socio-economic data. Thus we used spatial distribution data on population (POPs) and GDP (GDPs) to verify the feasibility of driving forces analysis method. We calculated mean values of POPs and GDPs with ArcGIS 10.3, named POPsm and GDPsm. Then, the relationship between POPsm and Pop was analyzed with SPSS (Table 1). The results indicated that Pop showed significant correlation with POPsm, and GDP showed extremely significant correlation with GDPsm. The results of correlation analysis indicated that socio-economic data of coastal cities can be used to conduct the analysis of driving forces.

Table 1 The relationship analysis between Pop and POPsm as well as GDP and GDPsm

Items

Pop

Items

GDP

POPsm

Pearson correlation

0.627*

GDPsm

Pearson correlation

0.886**

P value

0.022

P value

<0.01

2.      Why these seven factors were selected to investigate driving forces of landscape pattern changes?

Response 2: Climate change was one of the main driving forces of landscape changes[1]. Among the multiple climate change indicators, basic data of temperature and precipitation has the advantage of easily access, which is an important reason for the selection of the two factors.

During 2000-2015, wetlands were mainly converted into urban and farmland in Bohai coastal zone, indicating that urbanization and agricultural development were the main factors for the loss of wetland. Among the numerous urbanization indicators, economy and population were the most widely applied categories, given that they are the chief representatives of urbanization in China[2, 3]. GDP is a direct indicator of economic development. Thus, we chose GDP and population to explore the relationship between landscape changes and urbanization. The indicator of crop acreage (CA) and grain yield (GY) were the main matrices to represent agricultural development. The artificial reservoir/pond accounts for 77.27% of the total wetland ecosystem in 2000, the proportion increased to 84.53% in 2015. Most of the reservoir/pond was used to develop aquaculture for food and economic income. Thus, output of aquatic products (AQ) was chosen as a driver to explore the relationship between demand for aquatic products and changes in wetland landscape.

3. There are many spelling and unambiguous problems. Such as,1) line 43 in page 1, …and and urban; 2) line 18 in page 3, …of wetland area changes per year, why only used to wetland area changes;3) line 1 in page 4, periods: 2000–005? 4) In table 1, what is meaning of Pp/%? 5) In table2, the numbers should be reorganized, such as 1668.11, 1089.78. 6) In line 6 in page 8, what is F? 7) In line 24 in page 10, “Of”. 8) “5.2. Impacts of the Reduction of Wetland Ecosystem on Birds” is not directly related to the purpose of paper. 

Response 3: We are extremely grateful to you for pointing out these spelling mistakes and ambiguous problems. We have carefully examined the manuscript and corrected these mistakes where highlighted in blue. The modifications are as follows:

1)     We have corrected “…and and urban” (Page 1, Line 43) to “farmland and urban” (Page 1, Line 43);

2)     We have changed “wetland area changes” to “ecosystem area changes” (Line 2-3, Page 4);

3)     “periods: 2000–005” should be changed to “periods: 2000–2005”. But we have rewrote the paragraph about transition matrix (Page 4, Line 24-24 and Page 4, Line 1-2 of the original manuscript). The revised paragraph is on Page 4, Line 23-31.

4)     “Pp/%” in table 1(table 1 have been changed to table in revised version) means proportion, which is too long for the table to fit on the page. After modification and reorganization, table 1 have been changed to table 2 and “Pp/%” was replaced by “Proportion” to avoid ambiguity;

5)     We have reorganized the table, but table 1 have been changed to table 2 (Page 7);

6)     “F” means fragmentation. We have removed this index after discussion. Because NP and MPS can represent the fragmentation of the landscape.

7)     We have changed the title of section 5.2 to “Landscape Changes and Their Impact on Coastal Habitat” (Page 13).

8)      The title of section 5.2 had been changed to “Landscape Changes and Their Impact on Coastal Habitat”. The contents of this section have been modified and adjusted to be consistent with the manuscript. Line 30-35 (Page 13) and Line 46-48 (Page 13) in blue were the new added contents.

References:

1      Gao P, Niu X, Wang B, et al. Land use changes and its driving forces in hilly ecological restoration area based on gis and rs of northern china[J]. Scientific Reports. 2015, 5: 11038.

2      Su S, Xiao R, Jiang Z, et al. Characterizing landscape pattern and ecosystem service value changes for urbanization impacts at an eco-regional scale[J]. Applied Geography. 2012, 34: 295-305.

3      Chen J. Rapid urbanization in China: A real challenge to soil protection and food security[J]. Catena. 2007, 69(1): 1-15.

Round  2

Reviewer 3 Report

The authors do not answer my question. They just pressent the signifcance studying the landscape in the Bohai coastal zone. There are great number of such articles, what are the new methods and new results? And what are the implications of your study?

Page 5: Lines 7-22, Do you really want to make a good paper? Are these methods proposed by the authors themselves? If not, please cite references.

Figure 2: We cannot see any differences among these four maps, please present them more clearly. 

Page 8 Lines 4-29 poor expressions with many details, please present your major findings.

I do not think the conclusions are good enough.

Author Response

Comments and Suggestions for Authors

The authors do not answer my question. They just pressent the signifcance studying the landscape in the Bohai coastal zoneThere are great number of such articles, what are the new methods and new results? And what are the implications of your study?

We are grateful for your valuable comments and hard work concerning our manuscript entitled “Ecosystem Spatial Changes and Driving Forces in the Bohai Coastal Zone” (ijerph-418693). We have made some corrections and modifications which we hope meet your approval. We used a new indicator, landscape change index (LCI)[1] to depict the intensity of landscape changes (Page 4, Line 10-24). Landscape pattern and ecological process is closely related to ecosystem services, which have essential influence on the development of coastal communities. The analysis of past landscape changes is helpful for predicting future landscape changes and preserving important ecosystem services.  

The revised portion are marked in red in the manuscript. The review report and responses (in red) to your questions are listed below:   

Point 1: Page 5: Lines 7-22, Do you really want to make a good paper? Are these methods proposed by the authors themselves? If not, please cite references.

Response 1: We truly want to make a good paper and receive your approval. The methods depicted in Page 5, Line 7-22 was proposed by the Forest Science Department, Oregon State University, USA[2]. We are really sorry for our cursoriness. We have cited representative reference according to your suggestions. 

Point 2: Figure 2: We cannot see any differences among these four maps, please present them more clearly. 

Response 2: Since our study area is long and narrow, it is difficult to present these four maps clearly. For brevity and consistency, we have changed the previous colors of each ecosystem to bright and recognizable in the manuscript. We have divided the whole area into three parts and enlarged figures were presented in Figure 1. We compared landscape distribution in 2000 and 2015, significant change areas were marked in black circles. 

Point 3: Page 8 Lines 4-29 poor expressions with many details, please present your major findings.

Response 3: We have simplified the paragraphs in Page 8, Line 4-29. Significant changes mainly occurred in urban, farmland and wetland ecosystems. The trend of landscape changes is characterized by the expansion of urban and the shrinkage of farmland and wetland. There is no obvious trend in dynamic degree, so we have kept the explanation about the significant amplitude of the K value in the manuscript.     

Point 4: I do not think the conclusions are good enough.

Response 3: Thank you for your suggestions to improve the quality of the manuscript. The conclusions have been revised and divided into five paragraphs according to the results in section 4. Paragraph 1 (Page 14, Line 31-34) summarized the composition and area changes of each ecosystem; paragraph 2 (Page 14, Line 35-41) summarized the landscape change rate by quantifying the indicators of K and LCI; the landscape type transformation features were written in paragraph 3 (Page 15, Line 1-6). Paragraph 4 (Page 15, Line 7-8) and 5 (Page 15, Line 9-11) summarized the results landscape metrics and driving forces analysis. 

References:

1     Krajewski P, Solecka I, Mrozik K. Forest Landscape Change and Preliminary Study on Its Driving Forces in Ĺšlęża Landscape Park (Southwestern Poland) in 1883–2013[J]. Sustainability. 2018, 10(12): 4526.

2     Mcgarigal K M B. FRAGSTATS: spatial patterm analysis program for quantifying landscape structure[R]. Pacific Northwest Research Station, Porland, OR: USDA Forest Service General Technical Report PNW-GTR-351, 1995.

Reviewer 4 Report

The manuscript has been significantly improved. In section "5.3 Limitatios of Driving Forces Analysis" should added an explanation similar to Response 1.

Author Response

Comments and Suggestions for Authors

Point 1: The manuscript has been significantly improved. In section "5.3 Limitatios of Driving Forces Analysis" should added an explanation similar to Response 1.

Response 1: We are greatly thankful to you for your encouragement, valuable comments and hard work concerning our manuscript entitled “Ecosystem Spatial Changes and Driving Forces in the Bohai Coastal Zone” (ijerph-418693). It is of great significance for the improvement of the manuscript. We have added the explanation in section 5.3 according to your suggestions (Page 14, Line 6-16). The revised portion are marked in red in the manuscript.

Round  3

Reviewer 3 Report

Thanks the authors for their efforts. The manuscript requires thorough proofreading, not just about the language, but also about the content and logic.

Author Response

Point 1: Thanks the authors for their efforts. The manuscript requires thorough proofreading, not just about the language, but also about the content and logic.

Response 1: Thank you very much for your valuable suggestions and great efforts on our manuscript. The language of the manuscript was modified and polished by native English speakers. The structure and content of the manuscript was adjusted according to the suggestions of reviewers. We also proofread the article carefully. The revised portion are marked in red in the manuscript. 
